# Development of a method for calculating effective displacement damage doses in semiconductors and applications to space field

Yosuke Iwamoto *, Tatsuhiko Sato

Nuclear Science and Engineering Center, Japan Atomic Energy Agency, Ibaraki, Japan

* iwamoto.yosuke@jaea.go.jp

**Data Availability Statement:** All relevant data are within the manuscript.

**Funding:** This work was supported by Japan Society for the Promotion of Science (JSPS) Grant-in-Aid for Scientific Research (KAKENHI) Grant

## Abstract

The displacement damage dose (DDD) is a common index used to predict the life of semiconductor devices employed in space-based environments where they will be exposed to radiation. The DDD is commonly estimated from the non-ionizing energy loss based on the Norgett-Robinson-Torrens (NRT) model, although a new definition for a so-called effective DDD considers the molecular dynamic (MD) simulation with the amorphization in semiconductors. The present work developed a new model for calculating the conventional and effective DDD values for silicon carbide (SiC), indium arsenide (InAs), gallium arsenide (GaAs) and gallium nitride (GaN) semiconductors. This model was obtained by extending the displacement per atom tally implemented in the particle and heavy ion transport code system (PHITS). This new approach suggests that the effective DDD is higher than the conventional DDD for arsenic-based compounds due to the amorphization resulting from direct impacts, while this relationship is reversed for SiC because of recombination defects. In the case of SiC and GaN exposed to protons, the effective DDD/conventional DDD ratio decreases with proton energy. In contrast, for InAs and GaAs, this ratio increases to greater than 1 at proton energies up to 100 MeV and plateaus because the defect production efficiency, which is the ratio of the number of stable displacements at the end of collision cascade simulated by MD simulations to the number of defects calculated by NRT model, does not increase at damage energy values above 20 keV. The practical application of this model was demonstrated by calculating the effective DDD values for semiconductors sandwiched between a thin glass cover and an aluminum plate in a low-Earth orbit. The results indicated that the effective DDD could be dramatically reduced by increasing the glass cover thickness to 200 μm, thus confirming the importance of shielding semiconductor devices used in space. This improved PHITS technique is expected to assist in the design of semiconductors by allowing the effective DDD values for various semiconductors having complex geometries to be predicted in cosmic ray environments.

Number 19H02652. The funder had no role in study design, data collection and calculation, decision to publish, or preparation of the manuscript.

## Introduction

The development of space technology and the execution of space missions require materials that can maintain functional integrity under extreme conditions of heat, shock, and radiation [1]. There are challenges in materials science related to materials applications in space such as the design of metallic alloys for space exploration [2]. For semiconductors, high-energy particles (protons/ions) in space displace Si atoms from their proper positions in the crystal lattice, creating crystal defects that alter the electrical properties of the device and lead to circuit failure. This is a particularly critical problem in solar cell panels, as displacement damage causes a gradual loss of output power. Space missions will also require the use of devices that are not specifically designed for extraterrestrial environments. As an example, compound semiconductors such as silicon carbide (SiC), indium arsenide (InAs), gallium arsenide (GaAs) and gallium nitride (GaN) will be employed in various devices used in spacecraft. The displacement damage dose (DDD) is a parameter calculated as the product of the non-ionizing energy loss (NIEL) and the particle fluence and is used as an index to determine the lifetime of semiconductor devices when exposed to radiation such as might be encountered in space [3–9]. The GEANT4 [10] and FLUKA codes [11] can be used to calculate DDD values based on the NIEL database and the expected particle fluences. When employing these simulation codes, the NEIL database for various types of radiation obtained from the SR-NIEL calculator [12] is generally adopted. This database includes contributions from Coulomb scattering cross-sections, nuclear reaction models and neutron data processing such as performed using the NJOY code [13].

DDD values for semiconductors can be estimated using the displacement per atom (DPA) method. The DPA is the average number of displaced atoms per atom of a material based on the Norgett-Robinson-Torrens (NRT) model [14]. The DPA is widely used as an index of the displacement damage of materials exposed to radiation in nuclear power, fusion and accelerator facilities, and is expected to be proportional to the DDD value. The difference between the definitions of DDD and DPA is that DDD is based on damage energy whereas the DPA value is calculated using the number of Frenkel pairs of vacancies and interstitial atoms, equivalent to the ratio of the damage energy to the threshold energy of atomic displacement based on the NRT model. Over the last two decades, molecular dynamics (MD) simulations have been used to study defect formation in various materials. As an example, to consider the actual displacement damage, in 2018 Nordlund et al. reported that a lot of Frenkel pairs in metals recombine in a short time (about $10^{-10}$ seconds) calculated using MD simulations [15]. It is called the athermal-recombination-correction (arc). Here, we define the defect production efficiency as the ratio of the number of stable displacements at the end of collision cascade simulated by MD simulations to the number of defects calculated by NRT model. Gao et al. obtained the defect production efficiencies of compound semiconductors such as SiC [16], GaAs [17], GaN [18] and InAs [19] based on MD simulations and found that these efficiencies varied with the damage energies in the semiconductors. Thus, realistic estimations of radiation damage to semiconductors in space will require the calculation of the effective DDD ($DDD_{eff}$) values obtained by the product of $DDD_{conv}$ and the defect production efficiency, rather than the $DDD_{conv}$ values obtained from the SR-NIEL calculator. Our own group developed a method to calculate arc-DPA values, corresponding to the $DDD_{eff}$, for all metals and for incident particles such as neutrons, protons, electrons and heavy ions over a wide energy range. This method incorporates the results of the arc model while using Monte Carlo (MC) particle transport codes such as the particle and heavy-ion transport code system (PHITS) [20, 21]. In this manner, more realistic displacement damage values for metals were obtained compared with those generated using the conventional DPA process based on the NRT model (NRT-DPA), corresponding to the $DDD_{conv}$.

In the present work, we improved the arc-DPA calculation method in conjunction with the PHITS code to calculate $DDD_{eff}$ values for semiconductors, taking into account the defect production efficiencies of SiC, GaAs, InAs and GaN. To highlight the practical applications of this improved method, we calculated $DDD_{eff}$ values for 300 μm thick specimens of these semiconductors with a cover glass made of silicon dioxide and an aluminum base plate in low-Earth orbit (400 km) and in deep space using the cosmic ray source mode in the PHITS code. We assumed protons for Trapped Particles (TP) [22] and almost all charged particles for Galactic Cosmic Rays (GCR) [23] in low-Earth orbit, and charged particles for GCR in deep space.

## Calculation methods

The techniques used to calculate DDD values via the MC particle transport method were developed in the PHITS code. This section describes three aspects of these calculations: nuclear scattering and reactions, approximation of displacement damage, and implementation of defect production efficiencies, together with applications to space irradiation.

### Nuclear scattering and reactions

In calculations involving incident hadrons (protons and heavy ions) with energies greater than approximately 10 MeV/nucleon, the contributions of nuclear elastic and inelastic scatterings to the DDD were considered in the PHITS physics models. Details of the physics models in the PHITS code have been previously reported [24, 25].

### Approximation of displacement damage

In our previous studies, a DPA calculation method was developed and implemented in the PHITS code [24–27]. This process is briefly described here. The NRT-dpa cross-section in PHITS can be expressed based on the coulomb scattering cross-section as

$$\sigma_{\text{NRT-dpa}}(t) = \sum_i \int_{t_d}^{t_i^{\text{max}}} N_{\text{NRT-dpa}}(t_i) \frac{d\sigma_{\text{Coul}}(t_i)}{dt_i} dt_i \tag{1}$$

where $i$ is the type of charged particle, $t$ is a dimensionless collision parameter related to the recoil energy [28], $t_d$ is the dimensionless energy of the average threshold energy, $E_d$, $t^{\text{max}}$ is the dimensionless energy of the maximal transferred energy, and $\frac{d\sigma_{\text{Coul}}(t_i)}{dt_i}$ is the universal one-parameter differential scattering cross-section in reduced notation as reported by Lindhard et al. [28].

The differential scattering cross-section, $d\sigma/dt$, of charged particles is described based on classical scattering theory using the screening function $f(t^{1/2})$. The equation developed by Lindhard et al. referred to above can be written as

$$\frac{d\sigma_{\text{Coul}}(t)}{dt} = \frac{\pi a_{\text{TF}}^2}{2} \frac{f(t^{1/2})}{t^{3/2}} \tag{2}$$

where $a_{\text{TF}}$ is the screening distance. The screening function, $f(t^{1/2})$, can be generalized to provide a one parameter universal differential scattering cross-section equation for interatomic potentials such as screened and unscreened Coulomb potentials. The general form of this equation is

$$f(t^{1/2}) = \lambda t^{\frac{1}{2}-m}[1 + (2\lambda t^{1-m})^q]^{-1/q} \tag{3}$$

where $\lambda$, $m$ and $q$ are constants for each metal. $N_{\text{NRT-dpa}}$ in Eq 1 is the average number of defects based on the NRT-dpa model using the phenomenological approach. $N_{\text{NRT}}$ is a

function of the damage energy and is given as

$$N_{\mathrm{NRT}}(T_{\mathrm{d}}) = \begin{bmatrix} 0, & T_{\mathrm{d}} < E_{\mathrm{d}} \\ 1, & E_{\mathrm{d}} < T_{\mathrm{d}} < \dfrac{2E_{\mathrm{d}}}{0.8} \\ \dfrac{0.8T_{\mathrm{d}}}{2E_{\mathrm{d}}}, & \dfrac{2E_{\mathrm{d}}}{0.8} < T_{\mathrm{d}} \end{bmatrix} \qquad (4)$$

where $T_{\mathrm{d}}$ is the damage energy. This value is the energy available to generate atomic displacements by elastic collisions [14].

Based on the conventional DPA calculation method using NIEL values, the conventional NIEL, $NIEL_{\mathrm{conv}}$, value related to the NRT-dpa cross-section of a material can be written as

$$NIEL_{\mathrm{conv}}(t) = \rho \sum_i \int_{t_{\mathrm{d}}}^{t_i^{\max}} T_{\mathrm{d}}(t_i) \frac{d\sigma_{\mathrm{Coul}}(t_i)}{dt_i} dt_i = \rho \frac{2E_{\mathrm{d}}}{0.8} \sigma_{\mathrm{NRT}}(t) \qquad (5)$$

where $\rho$ is the atomic density of the semiconductor. To obtain a universal one parameter differential scattering cross-section, the Ziegler-Biersack-Littmark (ZBL) screening potential [29] ($\lambda = 5.01$, $m = 0.203$ and $q = 0.413$) used in the SR-NIEL calculator was employed in Eq 5. The relativistic and quantum mechanical cross-section derived by McKinley and Feshbach [30] was used for the coulomb scattering cross-section of electrons. In this manner, the PHITS model was able to calculate NIEL and DDD values on an event-by-event basis for all incident particles and target semiconductors without the NIEL database.

Fig 1 plots the NIEL values for the proton, neutron and electron irradiation of silicon as determined using the PHITS model and numerical data obtained from the SR-NIEL calculator [12]. The displacement energy of silicon was 21 eV [31]. The energy range of the neutrons in this work was from $10^{-4}$ MeV to 10 GeV, that of protons from $10^{-4}$ MeV to 10 GeV and that of electrons from $10^{-1}$ MeV to 200 MeV. This plot confirms that the NIEL values calculated using the PHITS model were in good agreement with the data obtained from the SR-calculator.

## Implementation of defect production efficiencies

When determining the NIELconv values for semiconductors as well as the NRT-dpa cross-sections, the number of defects produced by a recoil will be directly proportional to the recoil energy. Consequently, the NIELconv will be proportional to the number of defects produced by the irradiation. However, it has been recognized that the NIEL value is not proportional to the number of defects because of the non-linear processes that take place in semiconductors that are associated with the formation of amorphous pockets [16–19]. Therefore, it is important to adopt NIEL values for semiconductors using MD simulations with amorphization models; Gao et al. [16–19] adopted the direct impact/defect stimulation model [32] as an amorphization model. Note that although MD simulations use realistic damage models, the accuracy of MD simulations in semiconductors is not clear due to the lack of experimental data. Further measurements are needed to benchmark the simulation in the future.

Fig 2 plots defect production efficiencies as functions of the cascade damage energy for SiC [16], GaAs [17], GaN [18] and InAs [19] as calculated by MD simulations. In the case of dpa calculations involving metals, the arc model has been applied to the NRT-dpa technique and the defect production efficiency when fitted based on the results of MD simulations can be

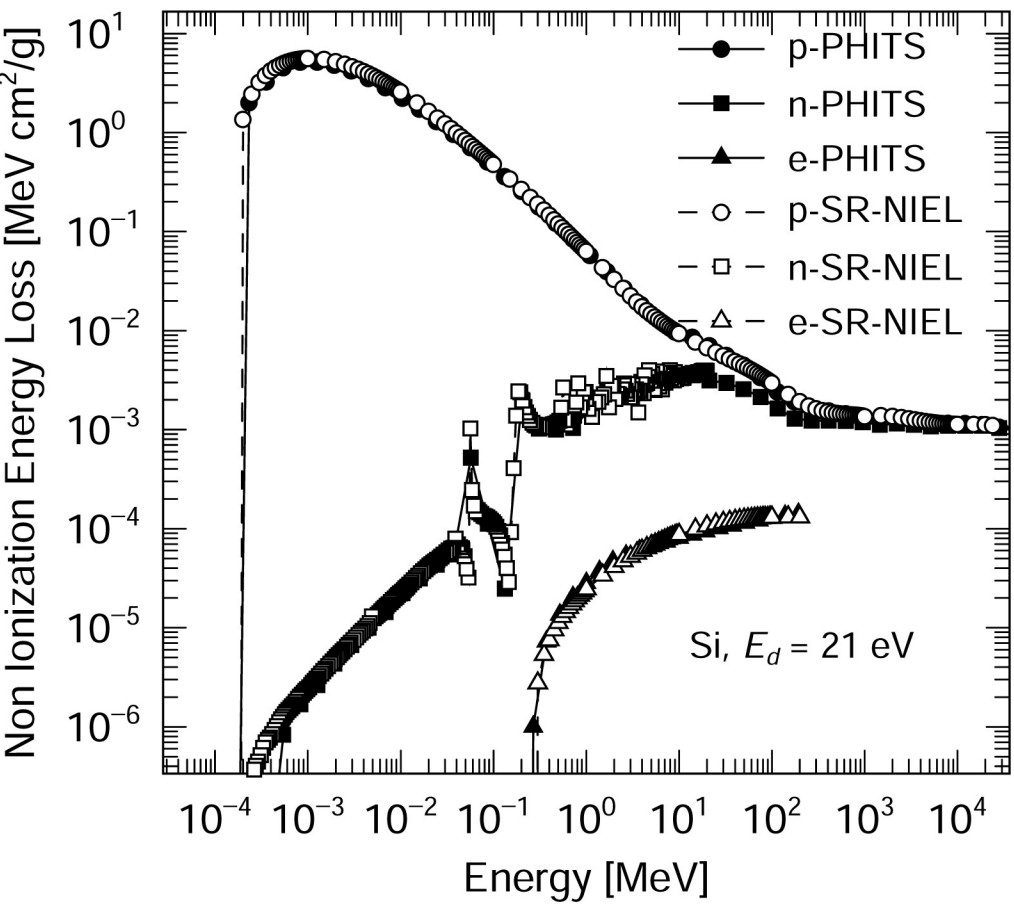

**Fig 1. NIEL values for the proton, neutron and electron irradiation of silicon.** This plot confirms that the NIEL values calculated using the PHITS model were in good agreement with the data obtained from the SR-calculator [12].

expressed as [15]

$$\kappa(T_\text{d}) = \frac{(a_\text{MD} - c_\text{MD})}{(2.5E_\text{d})^{b_\text{MD}}} T_\text{d}^{b_\text{MD}} + c_\text{MD} \tag{6}$$

where $k$ is the defect production efficiency as a function of the damage energy, $T_\text{d}$, and $a_\text{MD}$, $b_\text{MD}$, $c_\text{MD}$ and $E_\text{d}$ are constants that must be determined for a given material from MD simulations. In this work, the results calculated using an MD code for the semiconductors were fitted with Eq 6 to obtain values for $a_\text{MD}$, $b_\text{MD}$ and $c_\text{MD}$ (see Table 1). The effective NIEL value, NIEL$_\text{eff}$, was calculated as

$$NIEL_\text{eff}(t) = \rho \sum_i \int_{t_\text{d}}^{t_i^\text{max}} k(T_\text{d}) T_\text{d}(t_i) \frac{d\sigma_\text{Coul}(t_i)}{dt_i} dt_i = \rho \frac{2E_\text{d}}{0.8} \sigma_\text{arc}(t) \tag{7}$$

From Fig 2, it is evident that the GaAs and InAs data exhibit the same trend but the values for GaN and SiC show the opposite behavior. The surviving probabilities of defects for the arsenic compounds GaAs and InAs evidently increase along with the damage energy and this increase is nonlinear over the energy range from 1 to 20 keV. According to prior MD calculations [16–19], this nonlinear behavior can be explained by direct impact amorphization

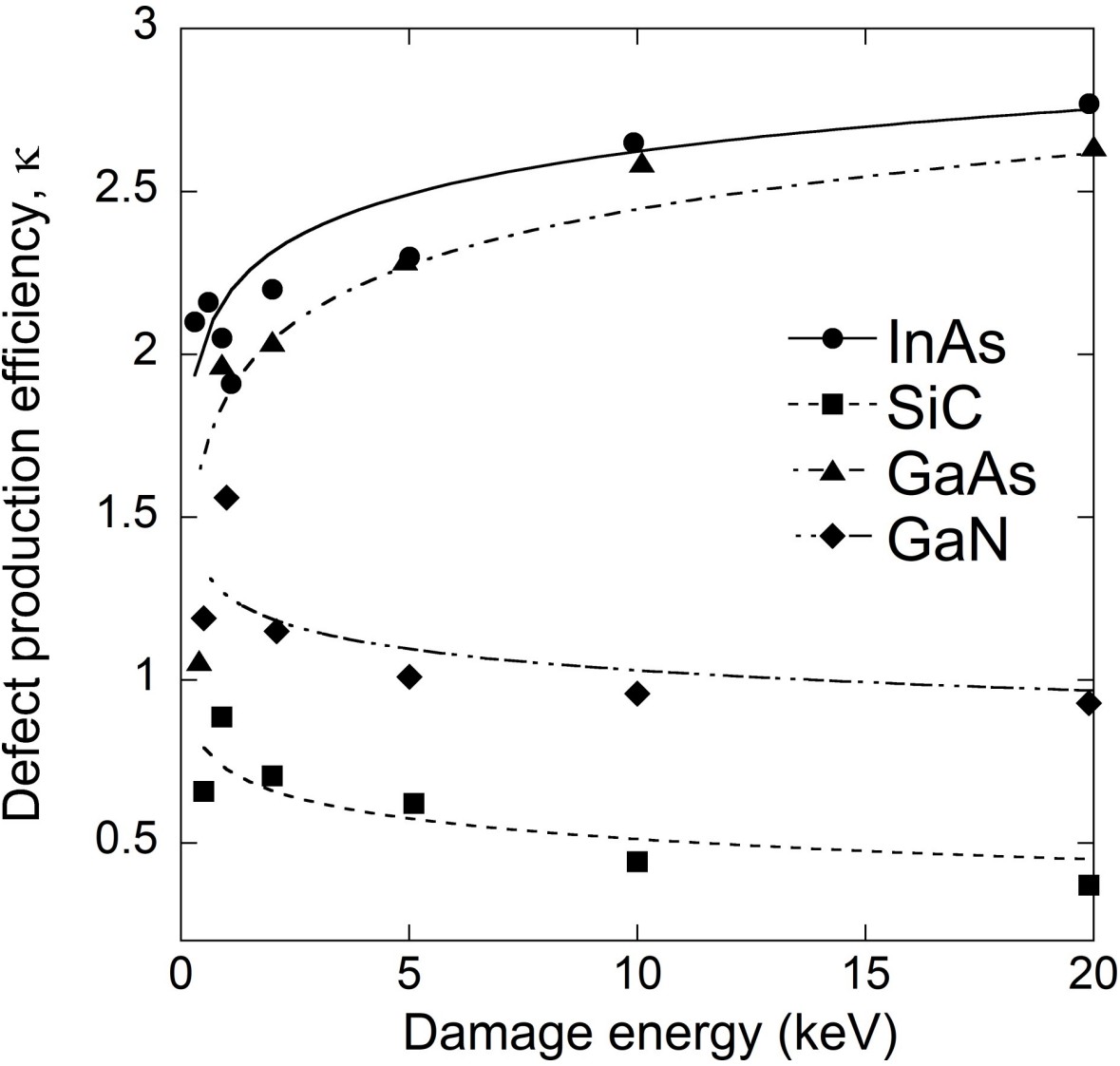

**Fig 2. Defect production efficiencies as functions of the cascade damage energy for SiC, GaAs, GaN and InAs.** The surviving probabilities of defects for the arsenic compounds GaAs and InAs evidently increase along with the damage energy and this increase is nonlinear over the energy range from 1 to 20 keV.

because the formation of these disordered regions will be exaggerated at higher energies for both InAs and GaAs. Conversely, the defect production efficiency decreases as the damage energy increases for GaN and SiC.

**Table 1. Material constants used for the calculation of damage production.** $E_d$ values for SiC [16], GaAs [17], GaN [18] and InAs [19] were obtained from the literature.

|  | SiC | InAs | GaAs | GaN |
|---|---|---|---|---|
| $E_d$ (eV) | 25 | 15 | 14 | 50 |
| $a_{MD}$ | 1.0 | 1.5 | 1.0 | 1.5 |
| $b_{MD}$ | -0.03 | -0.04 | -3 | -0.059 |
| $c_{MD}$ | 0.38 | 12 | 10 | 0.99 |

### Effects of irradiation on semiconductors in space

The displacement damage of semiconductors intended for used in solar panels in space was evaluated by determining DDD values for these materials in response to exposure to cosmic rays. These calculations employed the cosmic ray source mode in the PHITS code. This source mode reproduces the cosmic ray energy and angular distributions in space and in Earth's atmosphere based on four environments. These include GCR related to radiation in space [23], the atmosphere [32, 33], solar energetic particle (SEP) [34, 35] and TP [22] in low-Earth orbit. In this work, we calculated DDD values for semiconductors in low-Earth orbit (altitude of 400 km) using the GCR and TP fluxes as source terms and also in deep space using the GCR fluxes. The solar modulation potential (the so-called W-index) was set to zero to reproduce the solar minimum condition. Fig 3 shows the particle energy spectra of space radiation sources for a low-Earth orbit and for deep space as obtained using the cosmic ray source mode. The main radiation source in low Earth orbit is TP protons. The GCR flux includes almost all the charged particles found in space, including protons and He, Li, Be, B, C, N, O, F, Ne, Na, Mg, Al, Si, P, Cl, Ar, K, Ca, Sc, Ti, V, Cr, Mn, Fe, Co and Ni ions. Note that trapped electrons were not considered as a component of space radiation in this work because the electron NIEL value is smaller than that of protons.

Fig 4 shows the DDD geometry calculated for a semiconductor device exposed to cosmic rays in the PHITS code. In this scenario, the semiconductor is placed at the center of a sphere having a radius of 50 cm and irradiated with cosmic rays originating from the surface of the sphere. The device comprises a glass cover made of silicon dioxide ($SiO_2$), a semiconductor and an aluminum plate. The semiconductor is 300 μm thick while the aluminum plate has a thickness of 40 mm, which is sufficient to stop 100 MeV protons. The thickness of the glass cover was varied from 0 to 800 μm. The DDD value for InAs with a glass cover was also calculated to investigate the shielding of low-energy (below approximately 10 MeV) protons in a low-Earth orbit. The density of the glass cover was 2.5 $g/cm^3$ while that of the aluminum plate

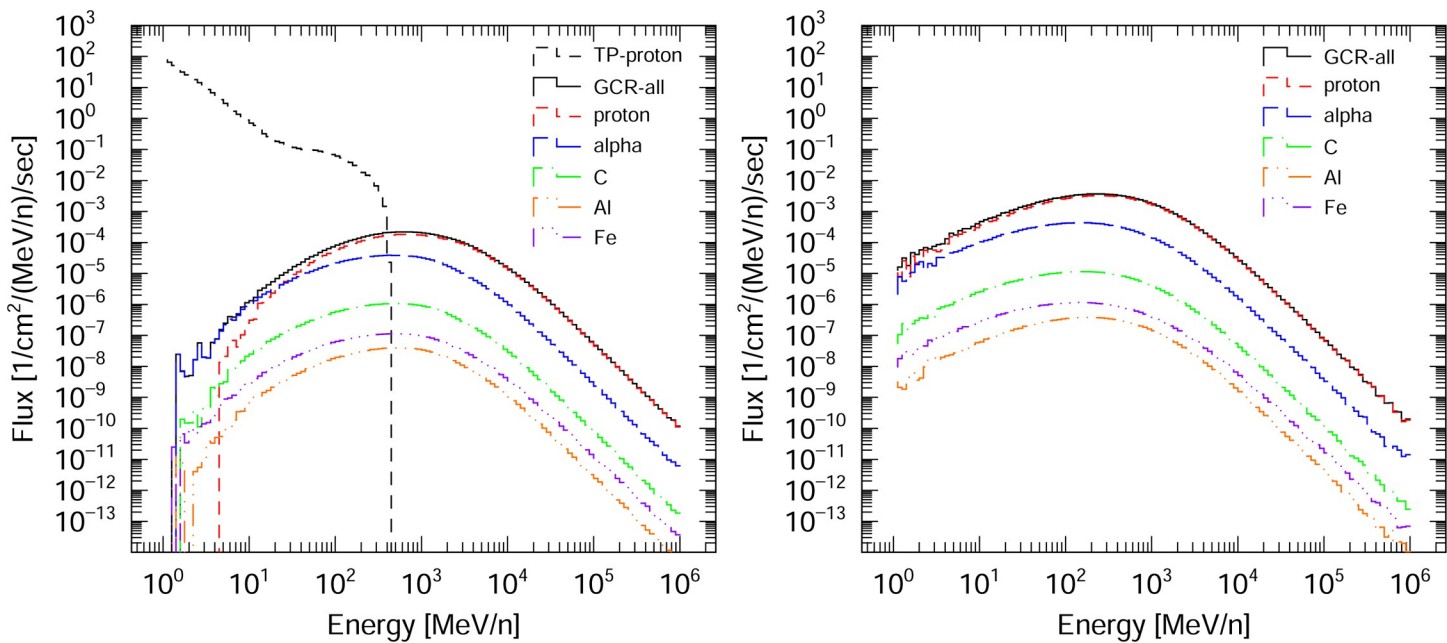

**Fig 3.** Particle energy spectra of radiation sources for a low-Earth orbit (left) and for deep space (right). The GCR and TP fluxes are included in low-Earth orbit and the GCR fluxes are included in deep space. Note that electrons in space-based radiation are ignored in the current version of PHITS.

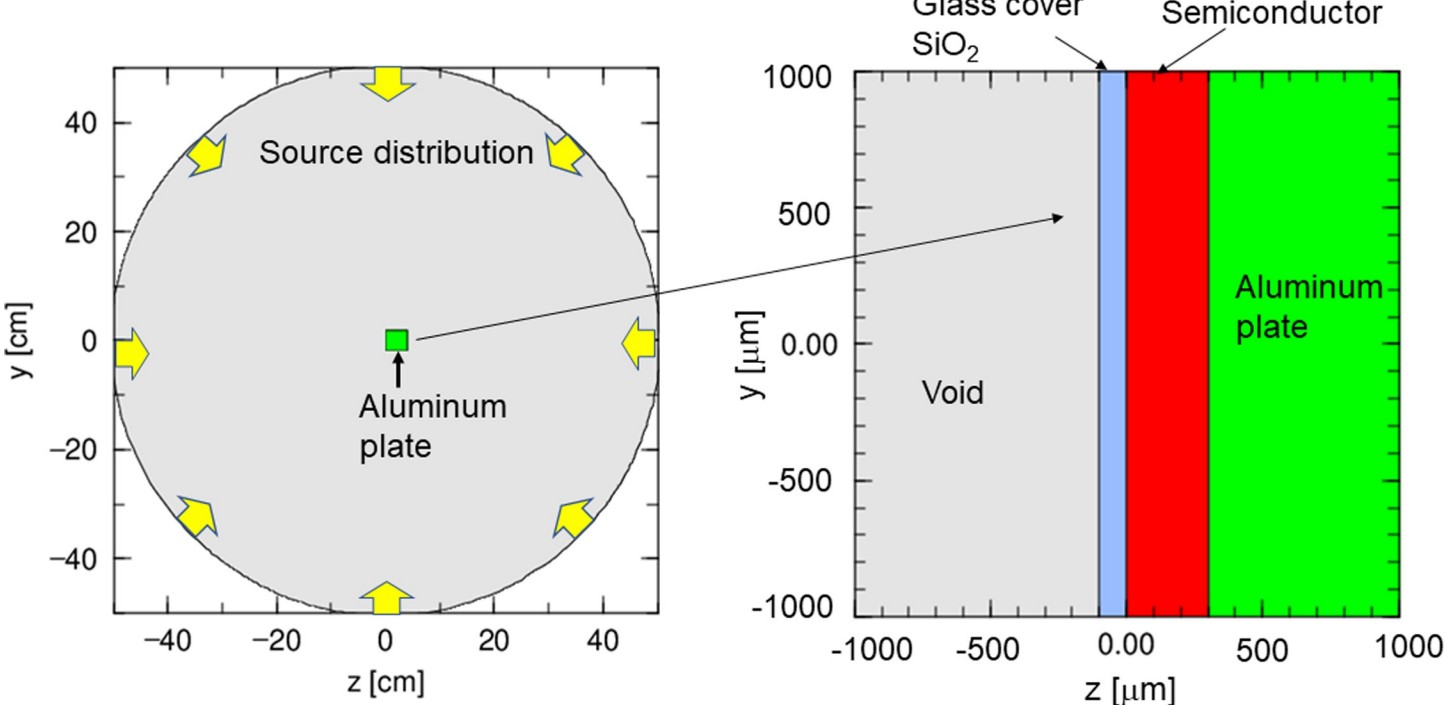

**Fig 4. Calculated DDD geometry for a semiconductor exposed to cosmic rays as determined in the PHITS code.** In this diagram, space-based radiation is emitted from the surface of the sphere towards the center. The device comprises a glass cover made of $SiO_2$, a semiconductor and an aluminum plate. The thickness of the glass cover was varied from 0 to 800 μm.

was 2.7 g/cm³. The densities of the semiconductors were 3.21 g/cm³ for SiC, 5.67 g/cm³ for InAs, 5.32 g/cm³ for GaAs and 6.15 g/cm³ for GaN.

## Results and discussion

### NIEL values

Fig 5 plots the $NIEL_{conv}$ values calculated using Eq 5 and the $NIEL_{eff}$ values calculated using Eq 7 for SiC, InAs, GaAs and GaN in the case of proton and neutron irradiation. These calculations used a neutron energy range of $10^{-11}$ MeV to 10 GeV and a proton energy range from $10^{-4}$ MeV to 10 GeV. In the case of the neutron-based NIEL values in the energy range below $10^{-4}$ MeV, the values for GaN are larger than those for GaAs, InAs and SiC by factors of 90, 50 and 845, respectively. In this neutron energy range, recoils produced by the neutron capture (n,γ) reaction are dominant for all nuclei. In addition, the $^{14}N(n,p)^{14}C$ nuclear reaction is also dominant for nitrogen. Fig 6 presents the recoil energy spectra based on the reactions of incident $10^{-11}$ MeV neutrons with $^{14}N$ as calculated using the PHITS code. This reaction produces not only $^{15}N$ atoms with energies below 5 keV via the $^{14}N(n,γ)$ reaction but also 30 keV $^{14}C$ atoms and 0.58 MeV protons via the $^{14}N(n, p)$ reaction. In the case of GaN, 30 keV $^{14}C$ atoms make the greatest contribution to the displacement damage. Therefore, GaN is the most sensitive to low-energy neutron irradiation environments (such as in spacecraft) among these semiconductors. In these environments, secondary neutrons are generated by neutron-based elastic and inelastic reactions between radiation and various materials. In the case of proton irradiation, Coulomb scattering between incident protons and the target is responsible for the

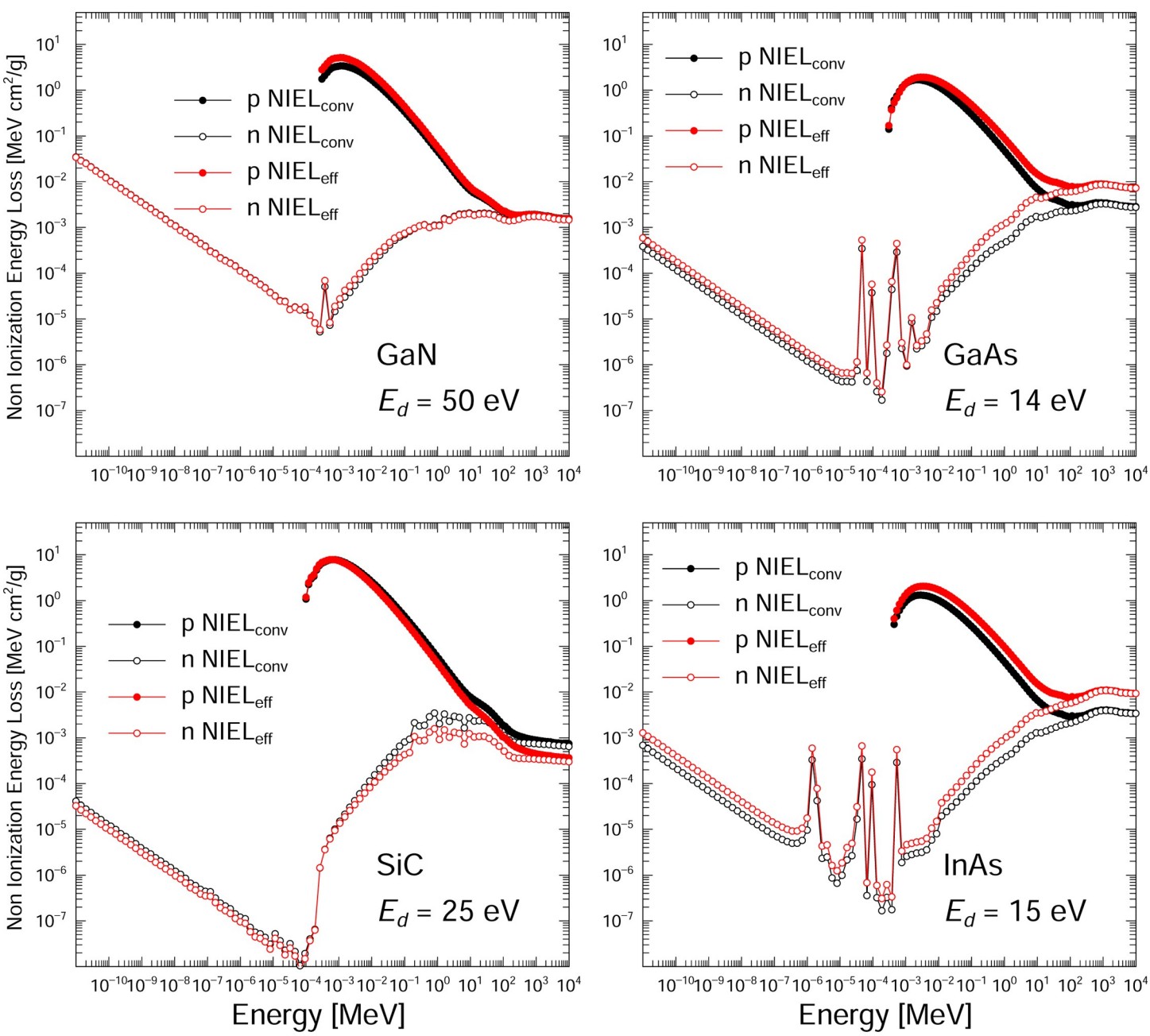

**Fig 5. NIEL$_{eff}$ and NIEL$_{conv}$ values for the proton and neutron irradiation of semiconductors as calculated using the PHITS code.**

majority of radiation-induced damage below 10 MeV while secondary particles produced by nuclear reactions are dominant in the high-energy range above 10 MeV.

Fig 7 shows the NIEL$_{eff}$ / NIEL$_{conv}$ ratios for SiC, InAs, GaAs and GaN in response to proton and neutron irradiation. With the exception of SiC, this ratio is greater than 1 over the entire energy range for both protons and neutrons because the ratio of the defect production efficiency to the damage energy remains higher than 1, as demonstrated in Fig 2. For SiC and GaN exposed to proton irradiation, this ratio decreases with increases in proton energy. In contrast, the results for InAs and GaAs indicate that the ratio increases along with the proton

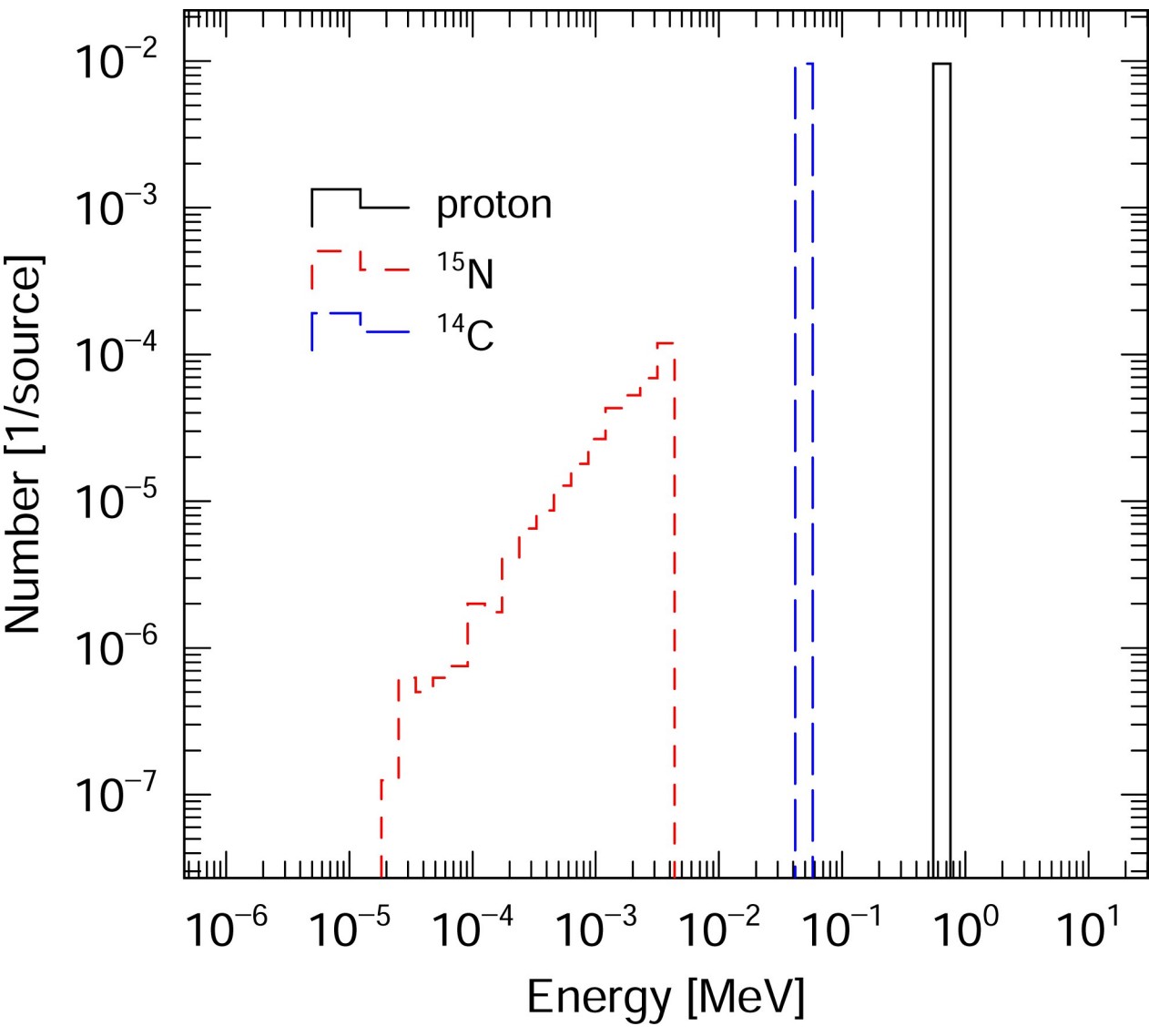

**Fig 6. Recoil energy spectra produced by the reaction of $10^{-11}$ MeV neutrons incident on $^{14}$N atoms.**

energy up to 100 MeV and then plateaus because the defect production efficiency at damage energies remained constant above 20 keV.

## DDD values of semiconductors in space

Table 2 summarizes the $DDD_{eff}$, $DDD_{conv}$ and $DDD_{eff}$ / $DDD_{conv}$ values for SiC, InAs, GaAs and GaN exposed to cosmic rays in low-Earth orbit and in deep space. Fig 8 provides a graph of the $DDD_{eff}$ and $DDD_{conv}$ data while Fig 9 shows the $DDD_{eff}$ / $DDD_{conv}$ ratios for these same materials.

A comparison of the $DDD_{eff}$ values associated with low-Earth orbit and deep space demonstrates that TP radiation in the former scenario makes the largest contribution to the displacement damage. This occurs because the NIEL value of protons increases with decreasing energy as a consequence of the significant contribution of Coulomb scattering between the proton and the target atom (see Fig 5). The ratio of $DDD_{eff}$ for low-Earth orbit to $DDD_{eff}$ for deep space was found to be 13.0 for SiC, 2.92 for InAs, 3.48 for GaAs and 4.43 for GaN.

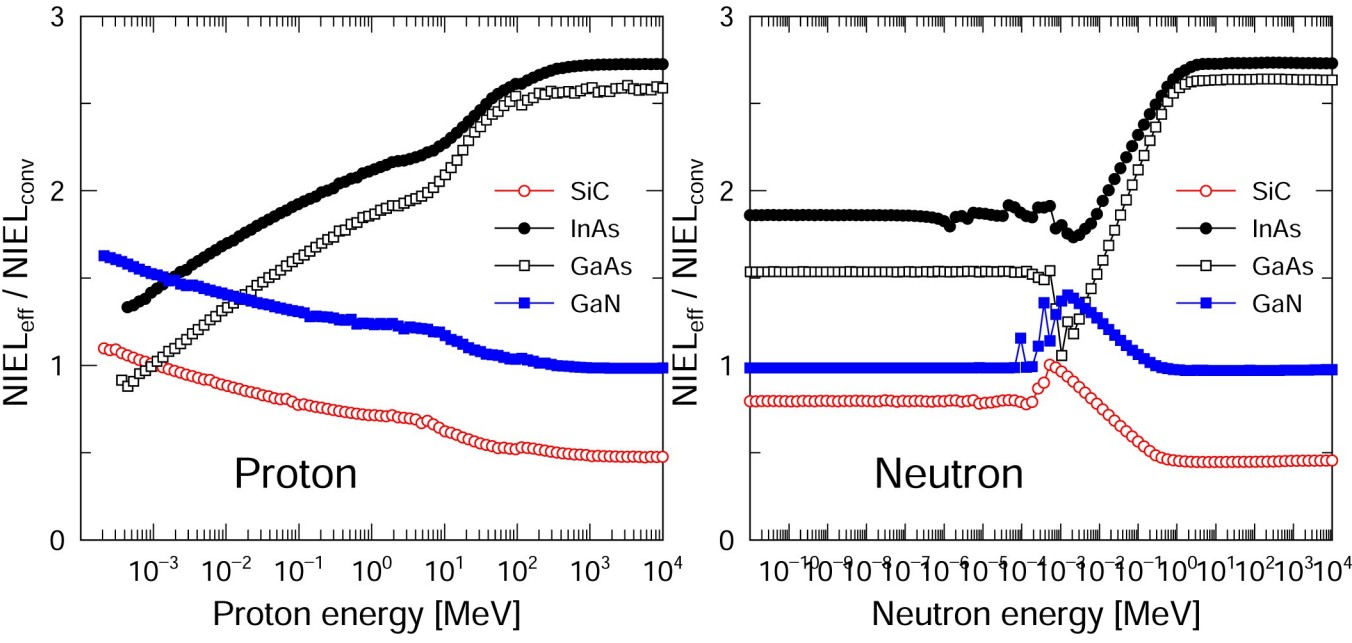

**Fig 7. NIEL$_{eff}$ / NIEL$_{conv}$ ratios for semiconductors in response to proton and neutron irradiation.**

Because the TP fluence for a low-Earth orbit, which is mainly composed of low-energy (below 10 MeV) protons, is about $10^4$–$10^5$ times higher than the GCR fluence (as shown in Fig 3), it is important to prevent low-energy protons from directly entering the semiconductor in low-Earth orbit. Therefore, a 100 μm thick glass cover can be used as a shielding material as shown in Fig 4. The radiation shielding ability of this cover was evaluated by calculating the DDD$_{eff}$ values for semiconductors in association with a TP radiation component while varying the cover thickness. Fig 10 plots the DDD$_{eff}$ values for InAs in conjunction with the TP component for a low-Earth orbit as a function of the glass cover thickness. The values are seen to be dramatically reduced as the cover thickness is increased to 200 μm and gradually decrease thereafter. The ratio of DDD$_{eff}$ with a thick glass cover to DDD$_{eff}$ without a glass cover was determined to be 0.35 for a 200 μm thickness and 0.26 for an 800 μm thickness. Calculations for the other semiconductors showed the same trend. Therefore, a glass cover with a thickness of approximately 200 μm is sufficient to decrease the effect of low-energy protons and so reduce displacement damage in semiconductors.

Fig 9 provides a comparison between DDD$_{eff}$ and DDD$_{conv}$ for a low-Earth orbit and for deep space. It is apparent that the DDD$_{eff}$ values for InAs and GaAs are higher than the DDD$_{conv}$ values by factors of 2.0–2.6. This effect is attributed to the increase in defect

**Table 2. DDD$_{eff}$, DDD$_{conv}$ and DDD$_{eff}$ / DDD$_{conv}$ for SiC, InAs, GaAs and GaN exposed to cosmic rays in low-Earth orbit and deep space.**

|  | Component | SiC | InAs | GaAs | GaN |
|---|---|---|---|---|---|
| **DDD$_{eff}$ (MeV/g/day)** | **Low-earth orbit** | 2.45E+03 | 9.31E+03 | 8.77E+03 | 3.54E+03 |
|  | **Deep space** | 1.89E+02 | 3.19E+03 | 2.52E+03 | 7.99E+02 |
| **DDD$_{conv}$ (MeV/g/day)** | **Low-earth orbit** | 3.52E+03 | 3.88E+03 | 4.04E+03 | 3.16E+03 |
|  | **Deep space** | 3.05E+02 | 1.30E+03 | 9.81E+02 | 7.61E+02 |
| **DDD$_{eff}$ / DDD$_{conv}$** | **Low-earth orbit** | 6.97E-01 | 2.40E+00 | 2.17E+00 | 1.12E+00 |
|  | **Deep space** | 6.19E-01 | 2.46E+00 | 2.57E+00 | 1.05E+00 |

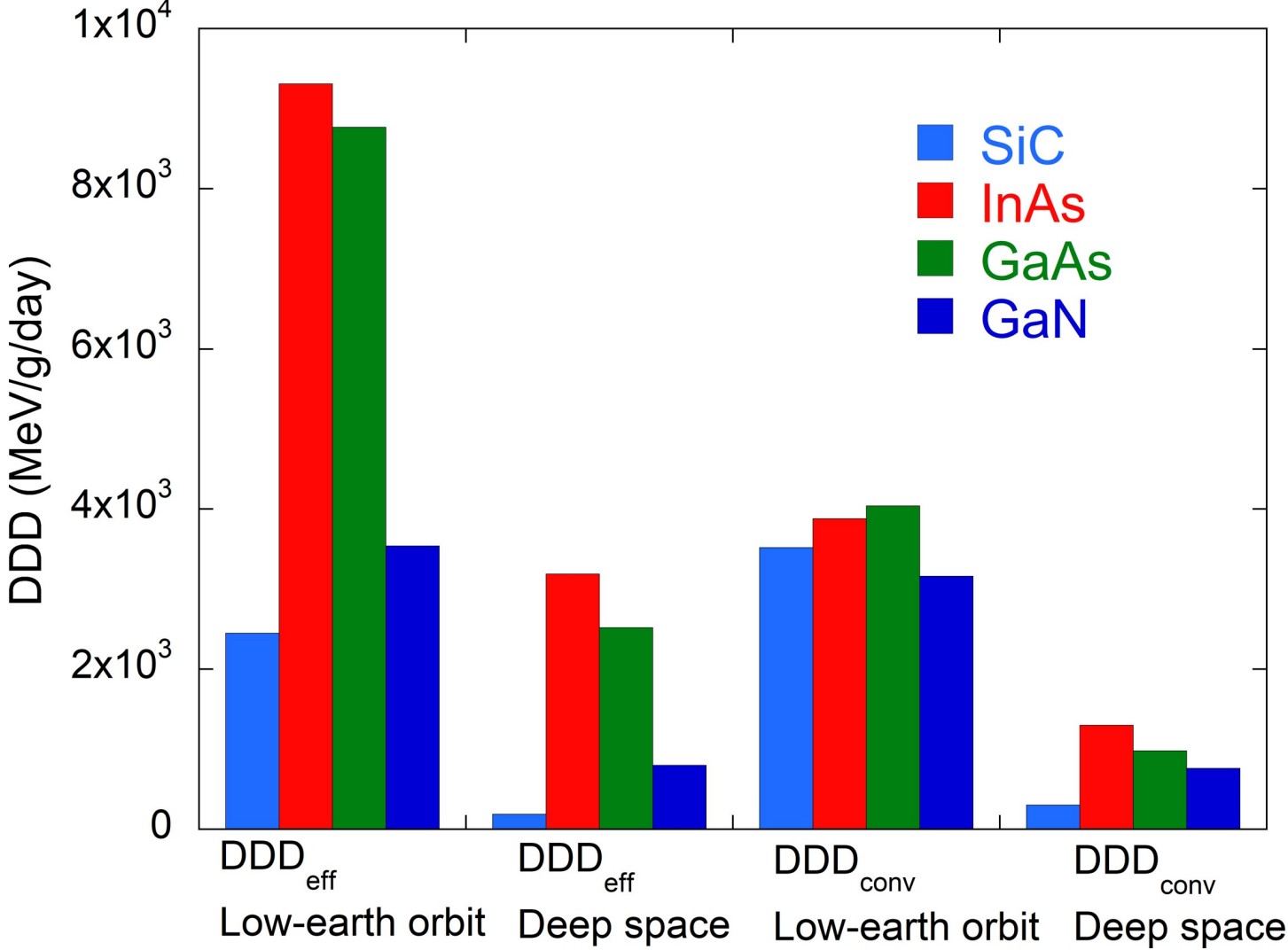

**Fig 8. DDD$_{eff}$ and DDD$_{conv}$ values for low-Earth orbit and deep space.**

production efficiencies with increases in the damage energy because of direct impact amorphization in the InAs and GaAs. In contrast, the DDD$_{eff}$ value for SiC was lower than DDD$_{conv}$ by a factor of 0.7 because the defect production efficiency decreased with increases in the damage energy based on the recombination of defects in this material. The DDD$_{conv}$ values calculated by the conventional method are quite different than the DDD$_{eff}$ values calculated based on a more realistic displacement damage effect, confirming the importance of estimating realistic displacement damage. As the development of semiconductors continues, it will be desirable to investigate the defect production efficiency of new semiconductors using MD simulations and to implement this effect in MC particle transport codes in future.

## Conclusion

The DDD is a common index used to determine the life time of semiconductor devices used in space radiation environments. DDD values are typically estimated from NIEL but a new definition of DDD, the so-called effective DDD, was recently proposed that takes into account the

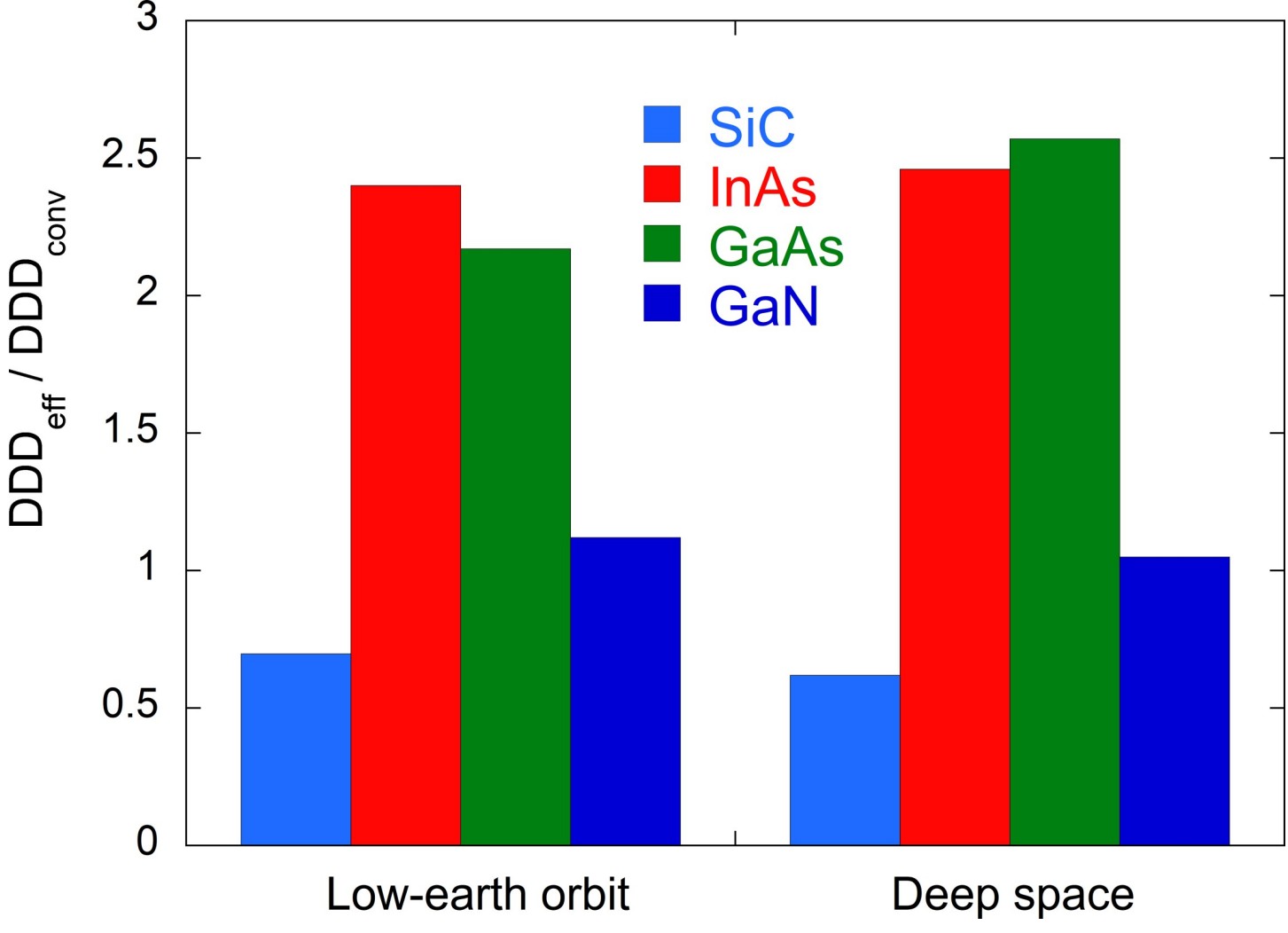

**Fig 9. DDD$_{eff}$ / DDD$_{conv}$ ratios for low-Earth orbit and deep space for SiC, InAs, GaAs and GaN.**

defect production efficiency obtained from MD simulations. The present work developed a new model for calculating both conventional and effective DDD values for SiC, InAs, GaAs and GaN semiconductors, by extending the DPA tally implemented in the PHITS code. The results suggest that the effective DDD will be higher than the conventional DDD for arsenic compounds as a consequence of amorphization resulting from direct impacts, while the opposite is true in the case of SiC because of recombination effects of defects. Data for SiC and GaN exposed to proton irradiation indicate that the ratio of the effective DDD to the conventional DDD decreases with proton energy. However, the results obtained for InAs and GaAs show that this ratio exceeds 1 for proton energies up to 100 MeV and eventually plateaus. This effect occurs because the defect production efficiency was constant at energies over 20 keV. The practical application of this approach was assessed by calculating the effective DDD values for these same semiconductors sandwiched between a thin glass cover and an aluminum plate and placed in a low-Earth orbit. This analysis established that the effective DDD could be dramatically reduced by increasing the thickness of the glass cover to 200 μm. This result confirms the importance of the shielding applied to semiconductor devices intended for use in space. The

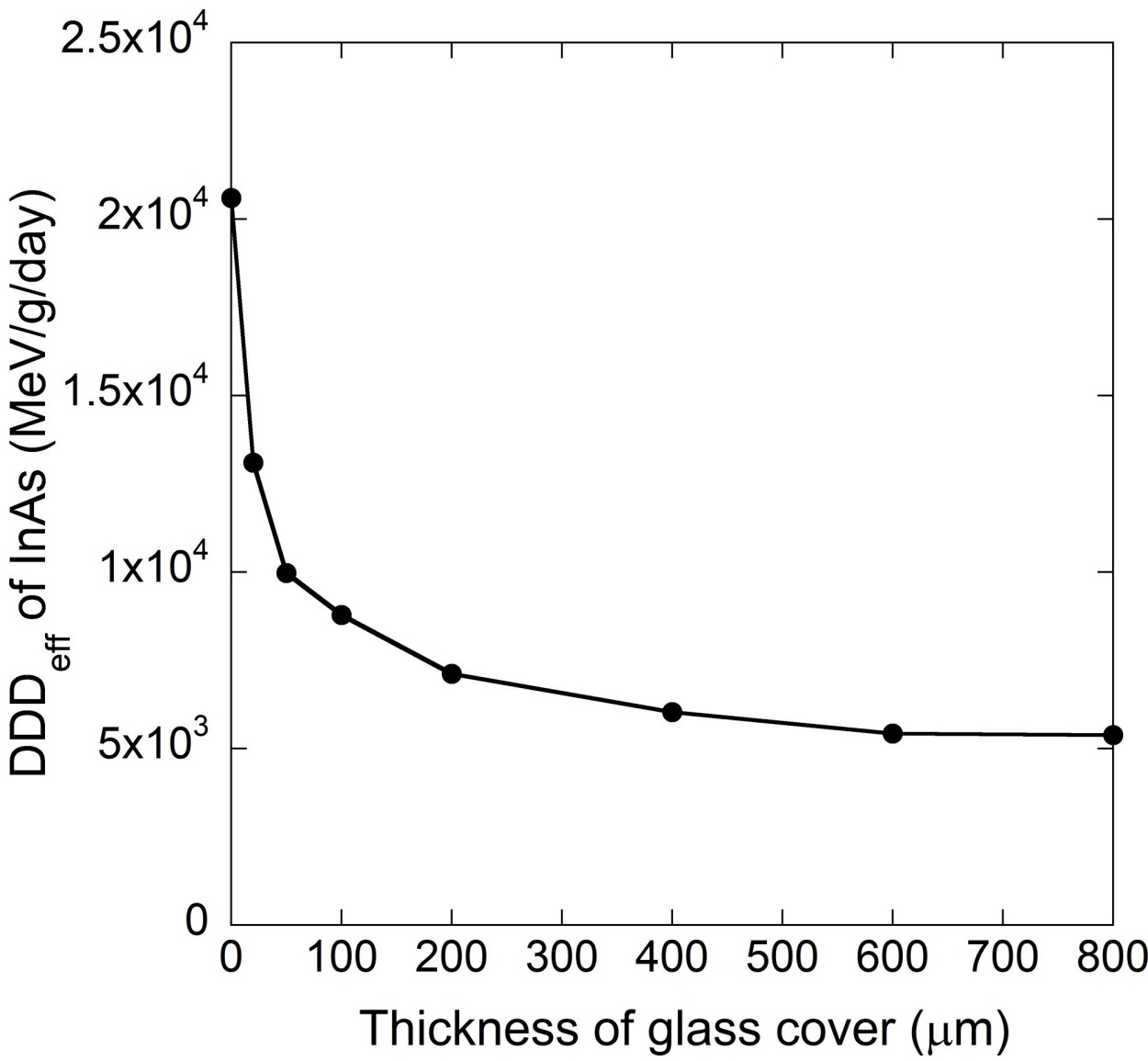

**Fig 10. DDD_eff values for InAs in a low-Earth orbit as a function of the thickness of a glass cover.** A glass cover with a thickness of approximately 200 μm is sufficient to decrease the effect of low-energy protons and so reduce displacement damage in semiconductors.

improved PHITS code is expected to contribute to the design of semiconductors in future by allowing effective DDD values to be calculated for various semiconductors having complex geometries and potential exposure to cosmic ray environments. In future work, we intend to include electrons in the space source function in the PHITS code and to calculate DDD values for electrons.

## Acknowledgments

We thank Edanz (https://jp.edanz.com/ac) for editing a draft of this manuscript.

## Author Contributions

**Conceptualization:** Yosuke Iwamoto.

**Data curation:** Yosuke Iwamoto.

**Formal analysis:** Yosuke Iwamoto.

**Funding acquisition:** Yosuke Iwamoto.

**Investigation:** Yosuke Iwamoto.

**Methodology:** Yosuke Iwamoto.

**Project administration:** Yosuke Iwamoto.

**Resources:** Yosuke Iwamoto.

**Software:** Yosuke Iwamoto.

**Supervision:** Tatsuhiko Sato.

**Validation:** Tatsuhiko Sato.

**Visualization:** Yosuke Iwamoto.

**Writing – original draft:** Yosuke Iwamoto, Tatsuhiko Sato.

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
