## [Decision Letter · Decision Letter 0]

25 Aug 2022

PONE-D-22-11567Development of a method for calculating effective displacement damage doses in semiconductors and applications to space fieldPLOS ONE

Dear Dr. Iwamoto,

Thank you for submitting your manuscript to PLOS ONE. After careful consideration, we feel that it has merit but does not fully meet PLOS ONE’s publication criteria as it currently stands. Therefore, we invite you to submit a revised version of the manuscript that addresses the points raised during the review process.

First of all congratulations to a very good written manuscript and sorry for the delay. I did my best to find reviewers after I have taken over the manuscript.There are some points raised by reviewer 2 which i would like you to point on:"Please define the term “defect production efficiency” in the introduction.Explain in clearer terms the difference between DDD and DDDeff. Deficiencies in the literature overview"Please consider to provide good, explanatory answers in the manuscript to support the understanding for future readers  

We look forward to receiving your revised manuscript.

Kind regards,

Bruno Merk

Academic Editor

PLOS ONE

Journal Requirements:

3. Please expand the acronym “JSPS KAKENHI” (as indicated in your financial disclosure) so that it states the name of your funders in full.

Reviewers' comments:

Reviewer's Responses to Questions

**Comments to the Author**

1. Is the manuscript technically sound, and do the data support the conclusions?

Reviewer #1: Yes

Reviewer #2: Yes

2. Has the statistical analysis been performed appropriately and rigorously? 

Reviewer #1: Yes

Reviewer #2: Yes

3. Have the authors made all data underlying the findings in their manuscript fully available?

Reviewer #1: Yes

Reviewer #2: Yes

4. Is the manuscript presented in an intelligible fashion and written in standard English?

Reviewer #1: Yes

Reviewer #2: Yes

5. Review Comments to the Author

Reviewer #1: The manuscript is very well written and presents a new development towards incorporating damage efficiencies into calculating DDD using fundamental understanding of existing damage models. Hence this work should be accepted in PLOS One. There are three comments i have for the authors and it would be nice if they could clarify these.

1. Damage efficiencies are dependent on crystallographic directions in single crystals since planer density of atoms along different crystallographic directions is different. We know that all semiconductors used in space tech are single crystals and hence would this be an important factor to consider in for example MD calculations. Something codes like DLPOLY and GULP are able to do while calculating Ed values for different materials.

2. Do some of the amorphization models consider single and double impact models developed by Gibbons, Ion Implantation in Semiconductors-Part I and Part II in Proceedings of IEEE 56 (1968) 295 and 60 (1972) 1062.

3. SiO2 is amorphous and i was wondering how do the authors define damage in an amorphous matrix to then quantify damage efficiency.

Reviewer #2: PONE-D-22-11567

Revision

Please define the term “defect production efficiency” in the introduction.

Explain in clearer terms the difference between DDD and DDDeff.

The introduction mentions the motivation of the work being evaluation of radiation damage semiconductors in space conditions, but the own irradiation environment of the space is not mentioned or defined. Challenges in materials science with respect to the application of materials in space are also not commented. Are the authors aware about recent developments in this area? Some references (non-exhaustive):

https://doi.org/10.1002/advs.202002397

https://doi.org/10.1038/s41563-018-0184-4

Lines 74-75: Confusing, please rewrite.

Please define T in lines 85-86.

Please define NIELconv in line 111.

Between lines 135 and 142, it has been said that NIEL is proportional to the number of defects produced by irradiation. Then the authors mention that this is not the case due to formation of amorphous pockets in semiconductors which are similar to defect production in metals. This is all confusing. Firstly, metals do not amorphize under irradiation (normally); I do not see such a correlation though. Secondly, NIEL is a parameter for nuclear collisions (i.e. non-ionizing). The creation of amorphous pockets in semiconductors within the nuclear regime is directly associated with the overlapping of damage cascades in the nuclear regime (see Gibbons model in his classical paper https://ieeexplore.ieee.org/abstract/document/1450784). I suggest rewriting this whole paragraph to streamline the ideas.

Corrections have been suggested based on MD simulations. However, how accurate are these MD simulations to allow such corrections in defect production efficiency?

6. PLOS authors have the option to publish the peer review history of their article (what does this mean?). If published, this will include your full peer review and any attached files.

Reviewer #1: No

Reviewer #2: No

---

## [Author Response · Author response to Decision Letter 0]

9 Sep 2022

Response to editor and reviewers

Editor comments:

First of all congratulations to a very good written manuscript and sorry for the delay. I did my best to find reviewers after I have taken over the manuscript.

There are some points raised by reviewer 2 which i would like you to point on:

"Please define the term “defect production efficiency” in the

introduction.

Explain in clearer terms the difference between DDD and DDDeff. 

Deficiencies in the literature overview"

Please consider to provide good, explanatory answers in the manuscript to support the understanding for future readers

We appreciate to find excellent reviewers, manage our paper and suggest valuable comments. We replied to all comments from you and reviewers. Especially, we modified the introduction a lot to support the understanding for readers. Please let us know if you have any comments.

 

Reviewer #1: The manuscript is very well written and presents a new development towards incorporating damage efficiencies into calculating DDD using fundamental understanding of existing damage models. Hence this work should be accepted in PLOS One. There are three comments i have for the authors and it would be nice if they could clarify these.

We appreciate the variable and useful comments for our revision of the paper. We modified the paper as reviewer #1 as following.

1. Damage efficiencies are dependent on crystallographic directions in single crystals since planer density of atoms along different crystallographic directions is different. We know that all semiconductors used in space tech are single crystals and hence would this be an important factor to consider in for example MD calculations. Something codes like DLPOLY and GULP are able to do while calculating Ed values for different materials.

The Damage efficienies described in this paper use the MDCASK code described in reference [12], which does not take into account crystallographic directions. However, it is noted that the calculated value of Ed is in good agreement with the calculated value of Ed considering crystallographic directions.

R. Devanathan and W.J. Weber, Journal of Nuclear Materials 278 (2000) 258-265.

2. Do some of the amorphization models consider single and double impact models developed by Gibbons, Ion Implantation in Semiconductors-Part I and Part II in Proceedings of IEEE 56 (1968) 295 and 60 (1972) 1062.

 It has been proposed that the amorphization behavior in SiC can be described by a direct-impact/defect-stimulated model [34], in which both amorphous nuclei (clusters) and defects that can stimulate amorphization are created directly within the cascade. 

No, it is not the Gibbons model, but the direct impact/defect stimulation model described in reference. This is also the amorphization model. Details are the following paper.

W.J. Weber, Models and mechanisms of irradiation-induced amorphization in ceramics, Nuclear Instruments and Methods in Physics Research B 166-167 (2000) 98-106.

We added the following sentence in the subsection, Implementation of defect production efficiencies. 

Gao et al adopted the direct impact/defect stimulation model as an amorphization model.

3. SiO2 is amorphous and i was wondering how do the authors define damage in an amorphous matrix to then quantify damage efficiency.

Since SiO2 is used as a radiation shielding material in this paper, the NIEL of SiO2 is not included in the calculations.

 

Reviewer #2: PONE-D-22-11567

Revision

We appreciate the variable and useful comments for our revision of the paper. We modified the paper as reviewer #2 as following.

Please define the term “defect production efficiency” in the introduction.

Explain in clearer terms the difference between DDD and DDDeff.

In the introduction, we defined the term “defect production efficiency” in the introduction and explain in clearer terms the difference between DDD and DDDeff. Below is a portion of the introduction.

Here, we define the defect production efficiency as the ratio of the number of stable displacements at the end of collision cascade simulated by MD simulations to the number of defects calculated by NRT model. Gao et al. obtained the defect production efficiencies of compound semiconductors such as SiC [16], GaAs [17], GaN [18] and InAs [19] based on MD simulations and found that these efficiencies varied with the damage energies in the semiconductors. Thus, realistic estimations of radiation damage to semiconductors in space will require the calculation of the effective DDD (DDDeff) values obtained by the product of DDDconv and the defect production efficiency, rather than the DDDconv values obtained from the SR-NIEL calculator.

The introduction mentions the motivation of the work being evaluation of radiation damage semiconductors in space conditions, but the own irradiation environment of the space is not mentioned or defined. 

In the introduction, we added the following information about the own irradiation environment in space.

We assumed protons for Trapped Particles (TP) and almost all charged particles for Galactic Cosmic Rays (GCR) in low-Earth orbit, and charged particles for GCR in the deep space.

Challenges in materials science with respect to the application of materials in space are also not commented. 

Are the authors aware about recent developments in this area? 

Some references (non-exhaustive):

https://doi.org/10.1002/advs.202002397

https://doi.org/10.1038/s41563-018-0184-4

We added the sentence about challenges in materials science at the beginning of the introduction. We also referred the following paper. 

Ghidini T. Materials for space exploration and settlement. Nature Mater. 2018;17: 846-850.

Tunes MA, Stemper L, Greaves G, Uggowitzer PJ, and Pogatscher S. Prototypic lightweight alloy design for stellar-radiation environments. Adv. Sci. 2020;7: 2002397.

Lines 74-75: Confusing, please rewrite.

In the revisions made prior to the submission of this paper, we forgot to delete this sentence. Therefore, it has been removed.

Please define T in lines 85-86.

T is the recoil energy. However, T was deleted because T is not used later in this paper.

In relation to T, Tmax was also deleted for the same reason.

Please define NIELconv in line 111.

Based on the conventional DPA calculation method using NIEL values, the NIELconv value related to the NRT-dpa cross-section of a material can be written as

->

Based on the conventional DPA calculation method using NIEL values, the conventional NIEL, NIELconv, value related to the NRT-dpa cross-section of a material can be written as

Between lines 135 and 142, 

it has been said that NIEL is proportional to the number of defects produced by irradiation. Then the authors mention that this is not the case due to formation of amorphous pockets in semiconductors which are similar to defect production in metals. This is all confusing. 

Firstly, metals do not amorphize under irradiation (normally); I do not see such a correlation though. Secondly, NIEL is a parameter for nuclear collisions (i.e. non-ionizing). 

The creation of amorphous pockets in semiconductors within the nuclear regime is directly associated with the overlapping of damage cascades in the nuclear regime (see Gibbons model in his classical paper https://ieeexplore.ieee.org/abstract/document/1450784). 

I suggest rewriting this whole paragraph to streamline the ideas.

We apologize for any confusion our incorrect statement may have caused you.

>Firstly, metals do not amorphize under irradiation (normally).

Yes. I agree that metals do not amorphize under irradiation. 

>Secondly, NIEL is a parameter for nuclear collisions (i.e. non-ionizing). 

>The creation of amorphous pockets in semiconductors within the nuclear regime is directly >associated with the overlapping of damage cascades in the nuclear regime

The amorphization model of semiconductors were included in MD simulations by Gao et al. and overlapping of damage cascade in semiconductors. The amorphization model in MD simulation by Gao et al. is not Gibbons model but the direct impact/defect stimulation model described in reference. Details are the following paper. 

W.J. Weber, Models and mechanisms of irradiation-induced amorphization in ceramics, Nuclear Instruments and Methods in Physics Research B 166-167 (2000) 98-106.

We rewrote this whole paragraph to streamline the ideas as follows on page 6.

When determining the NIELconv values for semiconductors as well as the NRT-dpa cross-sections, the number of defects produced by a recoil will be directly proportional to the recoil energy. Consequently, the NIELconv will be proportional to the number of defects produced by the irradiation. However, it has been recognized that the NIEL value is not proportional to the number of defects because of the non-linear processes that take place in semiconductors that are associated with the formation of amorphous pockets. Therefore, it is important to adopt NIEL values for semiconductors using MD simulations with amorphization models; Gao et al. adopted the direct impact/defect stimulation model as an amorphization model. Note that although MD simulations use realistic damage models, the accuracy of MD simulations in semiconductors is not clear due to the lack of experimental data. Further measurements are needed to benchmark the simulation in the future.

Corrections have been suggested based on MD simulations. However, how accurate are these MD simulations to allow such corrections in defect production efficiency?

According to the papers by Gao et al, there is no information about the accuracy due to the lack of experimental data. We added the following sentence in the subsection “implementation of defect production efficiencies” on page 6.

Note that although MD simulations use realistic damage models, the accuracy of MD simulations in semiconductors is not clear due to the lack of experimental data. Further measurements are needed to benchmark the simulation in the future.

---

## [Decision Letter · Decision Letter 1]

6 Oct 2022

Development of a method for calculating effective displacement damage doses in semiconductors and applications to space field

PONE-D-22-11567R1

Dear Dr. Iwamoto,

We’re pleased to inform you that your manuscript has been judged scientifically suitable for publication and will be formally accepted for publication once it meets all outstanding technical requirements.

Kind regards,

Bruno Merk

Academic Editor

PLOS ONE

Additional Editor Comments (optional):

Congratulations, well done!

Reviewers' comments:

Reviewer's Responses to Questions

**Comments to the Author**

1. If the authors have adequately addressed your comments raised in a previous round of review and you feel that this manuscript is now acceptable for publication, you may indicate that here to bypass the “Comments to the Author” section, enter your conflict of interest statement in the “Confidential to Editor” section, and submit your "Accept" recommendation.

Reviewer #2: All comments have been addressed

2. Is the manuscript technically sound, and do the data support the conclusions?

Reviewer #2: Yes

3. Has the statistical analysis been performed appropriately and rigorously? 

Reviewer #2: Yes

4. Have the authors made all data underlying the findings in their manuscript fully available?

Reviewer #2: Yes

5. Is the manuscript presented in an intelligible fashion and written in standard English?

Reviewer #2: No

6. Review Comments to the Author

Reviewer #2: Comments have appropriately addressed. Revised paper can be now accepted. All the questions have been answered fully.

7. PLOS authors have the option to publish the peer review history of their article (what does this mean?). If published, this will include your full peer review and any attached files.

Reviewer #2: No

---

## [Editor Report · Acceptance letter]

11 Oct 2022

PONE-D-22-11567R1 

Development of a method for calculating effective displacement damage doses in semiconductors and applications to space field 

Dear Dr. Iwamoto:

I'm pleased to inform you that your manuscript has been deemed suitable for publication in PLOS ONE. Congratulations! Your manuscript is now with our production department. 

Kind regards, 

on behalf of

Prof. Dr. Bruno Merk 

Academic Editor

PLOS ONE